# Detection of *CTNNB1* Hotspot Mutations in Cell-Free DNA from the Urine of Hepatocellular Carcinoma Patients

**DOI:** 10.3390/diagnostics11081475

**Published:** 2021-08-14

**Authors:** Selena Y. Lin, Ting-Tsung Chang, Jamin D. Steffen, Sitong Chen, Surbhi Jain, Wei Song, Yih-Jyh Lin, Ying-Hsiu Su

**Affiliations:** 1JBS Science, Inc., Doylestown, PA 18902, USA; slin@jbs-science.com (S.Y.L.); jammind3@gmail.com (J.D.S.); sitongchen@gmail.com (S.C.); sjain@jbs-science.com (S.J.); fsong@jbs-science.com (W.S.); 2Department of Internal Medicine, National Cheng Kung University Hospital, College of Medicine, National Cheng Kung University, Tainan 701, Taiwan; ttchang@mail.ncku.edu.tw; 3Department of Surgery, National Cheng Kung University Medical College and Hospital, Tainan 701, Taiwan; 4The Baruch S. Blumberg Research Institute, Doylestown, PA 18902, USA

**Keywords:** hepatocellular carcinoma, cell-free DNA, beta-catenin, urine, mutation, recurrence

## Abstract

Hepatocellular carcinoma (HCC) is a leading cause of cancer-related deaths worldwide. The beta-catenin gene, *CTNNB1,* is among the most frequently mutated in HCC tissues. However, mutational analysis of HCC tumors is hampered by the difficulty of obtaining tissue samples using traditional biopsy. Here, we explored the feasibility of detecting tumor-derived *CTNNB1* mutations in cell-free DNA (cfDNA) extracted from the urine of HCC patients. Using a short amplicon qPCR assay targeting HCC mutational hotspot *CTNNB1* codons 32–37 (exon 3), we detected *CTNNB1* mutations in 25% (18/73) of HCC tissues and 24% (15/62) of pre-operative HCC urine samples in two independent cohorts. Among the CTNNB1-mutation-positive patients with available matched pre- and post-operative urine (*n* = 13), nine showed apparent elimination (*n* = 7) or severalfold reduction (*n* = 2) of the mutation in urine following tumor resection. Four of the seven patients with no detectable mutations in postoperative urine remained recurrence-free within five years after surgery. In contrast, all six patients with mutation-positive in post-operative urine recurred, including the two with reduced mutation levels. This is the first report of association between the presence of *CTNNB1* mutations in pre- and post-operative urine cfDNA and HCC recurrence with implications for minimum residual disease detection.

## 1. Introduction

Hepatocellular carcinoma (HCC) remains a leading cause of cancer-related deaths worldwide. The presence of HCC tumor-derived mutations in urine cell-free DNA (cfDNA) can serve as a non-invasive biomarker for disease detection and precision management [1,2]. Ample evidence exists that the Wnt/beta-catenin pathway plays a pivotal role in HCC development [3,4]. In particular, numerous mutations in the beta-catenin gene, *CTNNB1*, have been associated with HCC [5,6]. Patients with mutations in *CTNNB1* and other Wnt/beta-catenin pathway genes typically show reduced responses to both kinase inhibitors [7] and immunotherapy [8]. Somatic *CTNNB1* variants have therefore been proposed as potential markers for HCC detection [9] and therapeutic response monitoring [10,11,12,13]. Noninvasive detection of *CTNNB1* variants in urine would greatly facilitate the development of a test for detection of minimum residual disease after treatment and for monitoring primary and recurrence HCC.

The annual HCC recurrence rate after surgery exceeds 10% and reaches 80% after five years [14,15,16]. Previous studies have shown that early recurrence of HCC is associated with low survival [17,18]. Thus, it is critical to identify patients at high risk of recurrence as early as possible. At present, no specific biochemical or genetic markers for HCC recurrence are in clinical use, with most postoperative patients monitored using serum alpha-fetoprotein (AFP) measurements and imaging tests such as computed tomography and MRI. Early detection of tumor-derived *CTNNB1* mutations in urine cfDNA requires an assay, not only with high sensitivity, but with short amplicon size, as we have demonstrated previously [19]. In this study, a short amplicon (53 bp) qPCR assay targeting HCC-associated hotspot *CTNNB1* exon 3 mutations in codon 32–37 was used for mutation detection in urine cfDNA from patients with HCC and to investigate the feasibility of using urine DNA markers as a prognostic marker for HCC recurrence and for detecting minimum residual disease after treatment. Our results demonstrate that the presence of detectable *CTNNB1* mutations in both pre- and post-operative urine could be an indication of the presence of minimum residual disease and poor prognosis. 

## 2. Materials and Methods

### 2.1. Construction of CTNNB1 S37C Plasmid

The *CTNNB1* exon 3 region (NCBI Genbank no. AY463360; nt 27,007–27,219) was amplified from Hep3B cell genomic DNA by PCR (forward primer: 5′-CTGATTTGATGGAGTTGG-3′; reverse primer: 5′-CTGATTTGATGGAGTTGG-3′), end-polished, and blunt-end-ligated into the vector using the Agilent PCR-Script Amp(+) Cloning kit (Agilent Technologies, Santa Clara, CA, USA). Site-directed mutagenesis was performed using forward (5′-CTGGAATCCATTGTGGTGCCACTAC-3′) and reverse (5′-GTAGTGGCACCACAATGGATTCCAG-3′) primers to generate the exon 3 S37C (hg19 chr3:41,266,113C>G) mutant plasmid, pCTNNB1-S37C. The plasmid sequence was verified by Sanger sequencing (data not shown).

### 2.2. Study Subjects and Samples

All samples, tissue (cohort 1) and urine (cohort 2), used in this study were obtained with informed consent from the National Cheng Kung University Hospital, Taiwan, in accordance with the guidelines of the Institutional Review Board. Information pertaining to each cohort is listed in Table 1 (cohort 1) and Table 2 (cohort 2). Note, cohort 1 and cohort 2 are two independent study populations. HCC is characterized by AJCC (TNM) staging and pathological grade 1 for well differentiated, grade 2 for moderate differentiated, grade 3 for poor differentiated, and grade 4 for undifferentiated as noted in each table.

### 2.3. Tissue DNA Isolation and Quantitation 

DNA from paraffin-embedded tissue sections was isolated using the MasterPure DNA kit (Epicenter, Madison, WI, USA), according to the manufacturer’s instructions. The concentration of liver tissue DNA was determined by a real-time PCR assay targeting the beta-globin gene, as previously described [20].

### 2.4. Urine Collection and DNA Isolation and Fractionation

Urine collection and DNA isolation were carried out as described previously [20]. Briefly, 0.5 M EDTA, pH 8.0, was added to a fresh urine sample to a final concentration of 10 mmol/L to inhibit possible nuclease activity, and the preserved sample was stored at −70 °C until DNA Isolation. 

To isolate total urine DNA, the frozen urine sample was thawed at room temperature and mixed with an equal volume of 6 mol/L guanidine thiocyanate by inverting the tube eight times. Then, 1 mL of resin (Wizard DNA purification kit; Promega, Madison, WI, USA) was added to the urine lysate, and the sample was incubated with gentle mixing for two hours to overnight at room temperature. The resin-DNA complex was pelleted by centrifugation, transferred to a mini-column (provided in the kit), and washed with a buffer provided by the manufacturer. The DNA was then eluted with Tris-EDTA buffer. DNA less than 1 kb, designated as low molecular weight (LMW) urine DNA, was obtained from total urine DNA using carboxylated magnetic beads (Beckman Coulter, Indianapolis, IN) as previously described [21]. 

### 2.5. Detection of CTNNB1 Codon 32–37 Hotspot Mutations

Mutations in codons 32–37 (hg19 chr3:41,266,097-41,266,114) of exon 3 of the *CTNNB1* gene were detected by the *CTNNB1* 32–37 mutation 53 bp qPCR assay developed by (JBS Science, Inc., Doylestown, PA, USA) according to the manufacturer’s protocol. The 18 bp *CTNNB1* BNA^NC^[NMe] clamp was purchased from Biosynthesis, Inc. (Lewisville, TX, USA). To evaluate assay performance, serial dilutions of pCTNNB1-S37C were used, ranging from 1 to 10,000 copies per reaction. 1500 genome copies of wild-type (WT) human genomic DNA (Roche Applied Science, Indianapolis, IN, USA) were used as negative controls. Standards were prepared by spiking mutant plasmid in a background of WT DNA.

### 2.6. Sanger Sequencing

Isolated tissue DNA (1 ng) was amplified in a PCR reaction using 0.5 µM primers (forward: 5′-CTGATTTGATGGAGTTGG-3′, reverse: 5′-GAGTGAAGGACTGAGAAAA-3′), 200 µM dNTPs, and HotStart Taq Plus polymerase (Qiagen, Valencia, CA, USA) in PCR buffer. The thermocycling program was as follows: 95 °C for 5 min to activate the polymerase, then 40 cycles at 95 °C for 30 s, 54 °C for 30 s, and 72 °C for 30 s, followed by a final extension at 72 °C for 4 min. 

For sequencing of PCR products generated in the presence of the BNA^NC^[NMe] clamp, tissue DNA (2 ng) was amplified in a PCR reaction using the primers contained in the *CTNNB1* 32–37 mutation qPCR assay along with the *CTNNB1* BNA^NC^[NMe] clamp. To increase the size of the amplified PCR product so that it is suitable for Sanger sequencing, a second round of PCR was performed using oligos containing an artificial tag sequence (forward: 5′-TCGTCGGCAGCGTCAGATGTGTATAAGAGACAG-3′; reverse: 5′-GTCTCGTGGGCTCGGAGATGTGTATAAGAGACAGCTGTGTGCTCTTCGTGTGTGGTG-3′). Sanger sequencing was performed in both directions (forward primer: 5′-TCGTCGGCAGCGTC-3′; reverse primer: 5′-GTCTCGTGGGCTCGGA-3′). All PCR products were purified using the Zymo DNA Cleanup and Concentration Kit (Zymo Research, Irvine, CA, USA) according to the manufacturer’s instructions, and sent to the NAPCore Facility (CHOP, Philadelphia, PA, USA) for sequencing. 

## 3. Results

### 3.1. Detection of CTNNB1 Hotspot Mutations in HCC and Non-HCC Tissues 

Literature analysis revealed that approximately 90% of *CTNNB1* mutations in HCC occur in two hotspot regions, codons 32–37 and 41–45, accounting for 55% and 34% of all *CTNNB1* mutations in HCC, respectively, as detailed in Appendix A and summarized in Figure 1. A short amplicon (53 bp) qPCR assay targeting the major hotspot region, encompassed by codons 32–37 was chosen for mutation detection in urine cfDNA. We first determined the sensitivity of the assay by using varying amounts of pCTNNB1-S37C plasmid, as detailed in Materials and Methods, spiked into a background of 1000 copies of sonicated WT human genomic DNA containing no mutations in the *CTNNB1* target region. The assay contained a sensitivity of 0.3% variant allele frequency (VAF) for *CTNNB1* codon 32–37 mutations and a linearity range of 3–10^4^ copies of mutated DNA (Appendix A). 

Next, we determined the accuracy of *CTNNB1* 32–37 qPCR assay by detecting mutations in liver tissue DNA from patients with hepatitis (*n* = 35), cirrhosis (*n* = 35), or HCC (*n* = 73) and validating by Sanger sequencing. Sociodemographic and clinicopathological characteristics of these patients are presented in Table 1. Of the 73 HCC samples tested, 24.6% (*n* = 18) contained a detectable mutation (Figure 2A), consistent with the reported detection rate of 20–25% [22,23]. HCC-adjacent tissues from the 18 *CTNNB1*-mutation-positive patients contained no detectable mutations, suggesting that the *CTNNB1* mutations detected in the tumor tissues were somatic rather than germline. None of the hepatitis or cirrhosis samples tested contained a detectable mutation at a limit of detection (LOD) of 10 copies/3 ng DNA. Thus, the *CTNNB1 32–37* hotspot mutation rate was significantly higher in HCC than non-HCC liver tissue (*p* < 0.001, Chi-square test).

By PCR-Sanger sequencing analysis in 5 HCC tissue samples with >15% *CTNNB1* 32–37 VAF, the *CTNNB1* mutations were verified in 3 of the 5 samples, A10K, A48K, and A38K (Figure 2B). To increase the sensitivity of the method, we re-sequenced the samples with the BNA^NC^[NMe] clamp (see Materials and Methods) included in the PCR amplification before sequencing. This approach confirmed the mutations in the remaining two samples, A64K and A65K (Figure 2C). We then randomly selected 23 HCC tissues negative for *CTNNB1* mutation for PCR-Sanger sequencing analysis. All 23 were found to contain only WT sequences in the region analyzed (Appendix A).

### 3.2. Detection of CTNNB1 Hotspot Mutations in Urine of HCC Patients

Upon validation of the *CTNNB1* 32–37 qPCR assay for detection of hotspot mutations in tissue DNA, we screened for *CTNNB1* 32–37 mutations in urine cfDNA of HCC patients. We screened a total of 62 HCC urine samples from a second, independent cohort (Table 2) by the *CTNNB1* assay and detected mutations in 24.1% (*n* = 15) of the patients. For 13 of these 15 patients, urine samples collected both before and after surgical resection of the tumor were available. Urine had been collected a day prior to surgery and then again at a follow-up visit (Table 3). Seven of the patients whose urine DNA contained *CTNNB1* mutations before surgery showed no detectable mutations after surgery. Among them was patient U13, who had the highest mutant copy number before surgery, indicating that the *CTNNB1* mutation detected in this patient’s preoperative urine was likely derived from the resected tumor. However, the other six patients remained positive for the *CTNNB1* mutation after surgery. Analysis of clinical follow-up records revealed that all 6 of these patients developed HCC recurrence within five years post-resection. One of these patients, U3, was diagnosed with lung metastasis slightly less than a year after surgery. Only four patients (31%, 4/13) were negative for recurrence at five years, and all four were also negative for *CTNNB1* hotspot mutations in postoperative urine. The overall sensitivity and specificity of the post-resection assay as a predictor of recurrence in this group of patients were 0.67 and 1.00, respectively. 

## 4. Discussion

Using a short-amplicon qPCR assay, we demonstrated for the first time, the detection of *CTNNB1* mutations in urine of HCC patients, and the prognostic utility of mutated *CTNNB1* in postoperative urine for HCC patients whose urine contained such mutations before surgery. Together, our findings, consistent with our previous reports [19,20,24,25] show that urine can serve as a source of tumor-derived cfDNA for noninvasive mutation detection in patients with HCC. In our patient cohorts, we detected *CTNNB1* mutations in tissue (25%) and urine (24%) samples at frequencies consistent with previous estimates of *CTNNB1* mutation frequency in HCC [26,27,28,29,30]. 

Compared with blood and tissue, the use of urine for detecting HCC mutations holds three major advantages: (1) urine is easy to collect in large volumes, (2) its collection does not require trained clinical personnel, and (3) the procedure is entirely noninvasive and can be repeated as frequently as necessary. However, compared with plasma cfDNA, urinary cfDNA is even more fragmented [31]. To overcome this, short-amplicon (<60 bp) PCR-based assays are needed for detection of specific mutations in highly fragmented cfDNA [19,20,32]. The ability to detect *CTNNB1* hotspot mutations in urine of HCC patients undergoing curative surgery may also contribute substantially to HCC precision medicine. *CTNNB1* has recently emerged as a potential biomarker to identify immunotherapy responders and non-responders [8,33]. The utility of detecting *CTNNB1* mutations in HCC therapy guidance remains to be evaluated.

Our study has several limitations. The number of HCC patients positive for *CTNNB1* mutations before resection for whom a postoperative urine sample was also available was small (*n* = 13). Nevertheless, six of these patients were shown to retain detectable levels of *CTNNB1* mutations after surgery, and all 6 developed HCC recurrence. However, 3 of the 7 patients negative for *CTNNB1* mutations in postoperative urine did recur within 5 years. These observations may indicate that even higher sensitivity is needed to detect the low postoperative levels of *CTNNB1* mutations in some HCC patients. Alternatively, new tumors may arise independently of the original malignancy, a frequently reported complication in patients with HCC [34,35]. Another limitation is that the assay does not distinguish among different mutations in the target region. It is therefore possible that some of the patients shown to be positive after surgery had a different variant prior to it. The possibility of the assay detecting some mutations adjacent to the codon 32–37 region also cannot be ruled out. 

Taken together, our results (1) demonstrate the feasibility of detection of HCC-derived mutations in the *CTNNB1* gene in urine cfDNA, (2) provide evidence of association between the presence of *CTNNB1* mutations in urine and the presence of minimum residual disease or HCC recurrence, and (3) warrant a further study for the application of urinary *CTNNB1* mutation analysis in HCC precision medicine/disease management. Larger, multi-center studies of the correlation between *CTNNB1* mutations in urine cfDNA and HCC tumor status are needed to evaluate the potential clinical utility of urinary *CTNNB1* mutation detection in liver cancer management and precision medicine.

## Figures and Tables

**Figure 1 diagnostics-11-01475-f001:**
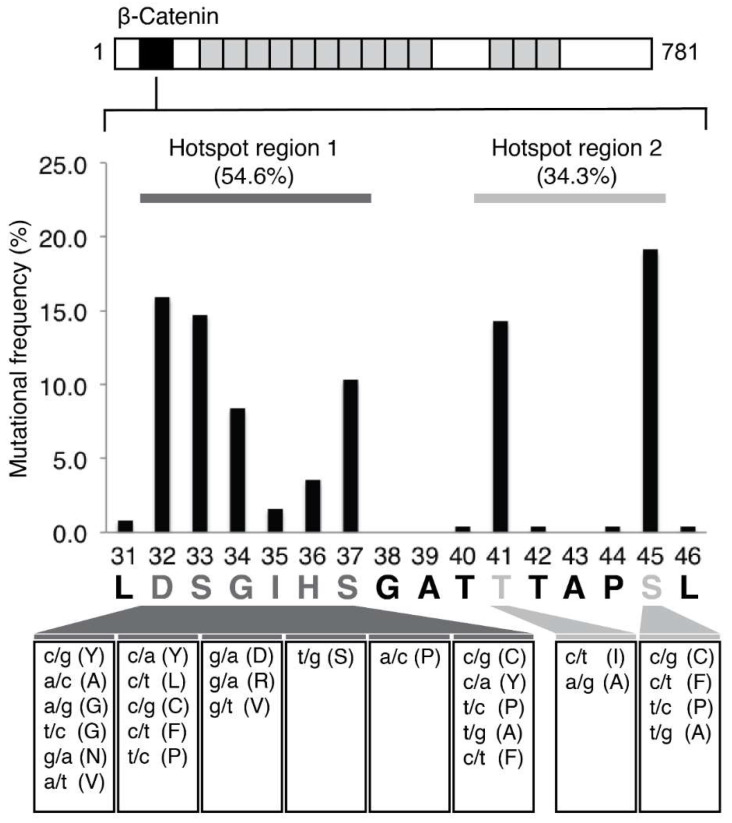
CTNNB1 exon 3 mutational frequency in HCC. Data from several studies that sequenced CTNNB1 exon 3 in patients with HCC were compiled (see Appendix A). In nearly 90% of all HCC tumors with a mutation in *CTNNB1*, the mutation resides within one of two hotspot regions: region 1 (codons 32–37; 54.6%) and region 2 (codons 41–45; 34.3%). The X-axis denotes the codon number and amino acid encoded by it. The box linked to an amino acid lists the reported missense mutations within the codon and the corresponding amino acid changes.

**Figure 2 diagnostics-11-01475-f002:**
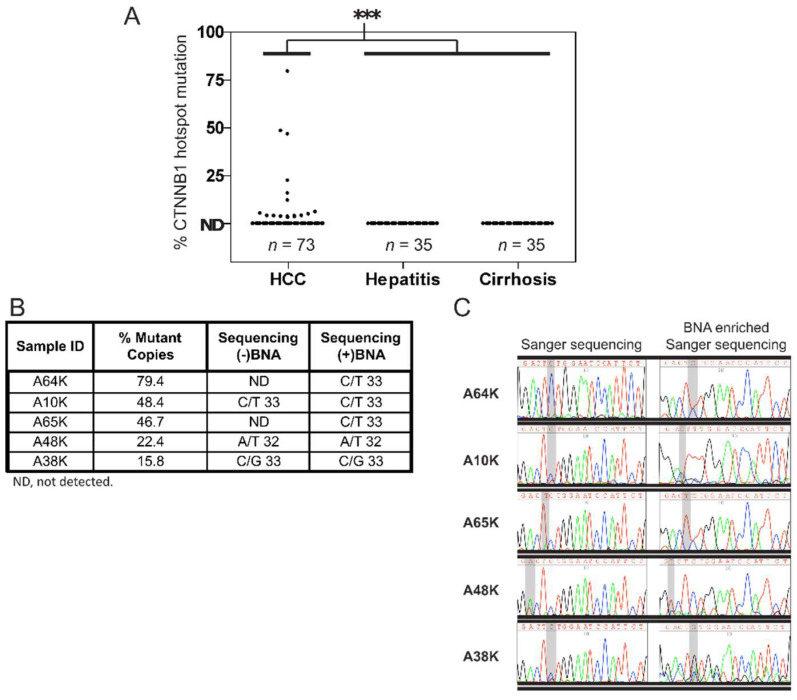
Analysis of *CTNNB1* mutations in tissue samples from patients with different liver diseases. (**A**) *CTNNB1* hotspot mutant allele frequencies obtained by the *CTNNB1* 32–37 mutation qPCR assay in liver tissues of patients with HCC, hepatitis, or cirrhosis. In the HCC subset (*n* = 73), 18 patients tested positive, while all of the hepatitis (*n* = 35) and cirrhosis (*n* = 35) samples tested negative. ***, *p* < 0.001. (**B**) Validation of the *CTNNB1* 32–37 mutation qPCR assay by Sanger sequencing. Five samples identified as positive by the qPCR assay with mutation frequencies of >15% were evaluated by Sanger sequencing. Of these five samples, three had detectable *CTNNB1* mutations. When Sanger sequencing was repeated after enrichment for the mutation using the BNA^NC^[NMe] in the PCR amplification reaction, *CTNNB1* mutations in the other two samples were also confirmed. The right panel shows Sanger sequencing chromatograms with and without BNA^NC^[NMe] enrichment. The mutation position is shaded in gray.

**Table 1 diagnostics-11-01475-t001:** Clinicopathological characteristics of the cohort 1 tissue DNA * analyzed in this study.

Diagnosis	Hepatitis (*n* = 35)	Cirrhosis (*n* = 35)	HCC (*n* = 73)
Mean age ± SD years	54 ± 12	56 ± 14	60 ± 12
Gender (Male:Female:Unknown)	17:18:0	23:12:0	45:20:8
Etiology			
HBV	3	6	31
HCV	22	17	19
HBV/HCV	9	12	2
Other	0	0	14
Unknown	1	0	7
Stage ^	NA	NA	
1			19
2			31
3			12
4			11
Grade ^#^	NA	NA	
1			9
2			41
3			15
Unknown			8
Tumor size, mean ± SD, cm	NA	NA	5.0±3.3
AFP level, ng/mL	NA	NA	
≤20			28
>20			37
Unknown			8

* Tissue DNA was isolated from tissue sections of the surgical resected FFPE tissue; ^, AJCC (TNM) staging, ^#^, pathological grading for differentiation; AFP, alpha-fetoprotein; HBV, hepatitis B virus; HCC, hepatocellular carcinoma; HCV, hepatitis C virus; SD, standard deviation; NA, not applicable.

**Table 2 diagnostics-11-01475-t002:** Clinicopathological characteristics of the cohort 2 patient urine samples analyzed in this study.

Diagnosis	HCC (*n* = 62)
Mean age ± SD years	59.9 ± 11.4
Gender (Male:Female)	44:18
Etiology	
HBV	30
HCV	15
HBV/HCV	1
Other	10
Unknown	6
Stage *	
1	18
2	28
3	12
4	2
Unknown	2
Grade ^#^	
1	8
2	38
3	14
Unknown	2
Tumor size, mean ± SD, cm	5.26 ± 3.27
AFP level, ng/mL	
≤20	38
>20	24
Unknown	0

* AJCC (TNM) staging; ^#^ pathological grading for differentiation; AFP, alpha-fetoprotein; HBV, hepatitis B virus; HCC, hepatocellular carcinoma; HCV, hepatitis C virus; SD, standard deviation.

**Table 3 diagnostics-11-01475-t003:** Detection of *CTNNB1* hotspot mutations in urine of HCC patients before and after tumor resection.

Sample ID	Serum AFP (ng/mL)	Tumor			*CTNNB1* 32–37 Mutation (Copies per mL of Urine) ^#^		Urine Collection Post-Tumor Resection (Months)	Recurrence ^^^	
		Stage *	Grade ^#^	Size (cm)	Before Resection	After Resection		Detected	Months Post-Resection
U1	6.9	1	G2	1.9	2–20	ND	10	No	NA
U2	5.0	1	G2	3.5	2–20	29	3	Yes	26
U3	19.1	2	G3	14.5	2–20	881	3	Lung metastasis	13
U4	1.8	3A	G2	8.0	2–20	23	2	Yes	51
U5	11.7	2	G3	4.4	2–20	2–20	2	Yes	21
U6	6.5	3A	G2	3.5	21	ND	2	No	NA
U7	NT	2	G2	1.5	24	ND	10	Yes	21
U8	3.8	1	G2	1.5	29	ND	2	No	NA
U9	6101.0	2	G2	3.0	39	ND	1	Yes	4
U10	4.0	1	G1	2.0	42	ND	3	No	NA
U11	<1.5	3C	G3	6.0	101	34	2	Yes	7
U12	NT	1	G2	4.0	142	26	1	Yes	47
U13	4.3	3A	G3	5.0	498	ND	1	Yes	57

^^^ Recurrence was monitored for five years after surgery and detected by CT scan or MRI; * AJCC (TNM) staging; ^#^ pathological grading for differentiation; NT, not tested; ND, not detected; NA, not applicable; ^#^ average of two experiments.

## Data Availability

All relevant data can be found in the manuscript and the Appendix A.

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
