# Peer review of "Detection of CTNNB1 Hotspot Mutations in Cell-Free DNA from the Urine of Hepatocellular Carcinoma Patients"

_diagnostics, 2021, doi:10.3390/diagnostics11081475_

Round 1
Reviewer 1 Report
Thank you for the opportunity to review the manuscript “Detection of CTNNB1 hotspot mutations in urine cell-free DNA of hepatocellular carcinoma patients” by Lin et al. In this study, the authors analyze pre- and post-operative urine for presence of a CTNNB1 hotspot mutation as a measure of residual disease and HCC recurrence risk following resection.
The authors show the qPCR approach can detect CTNNB1 32-37 mutations specifically in HCC tissue relative to tumor-adjacent, cirrhotic, and inflamed tissue as well as verify the specific CTNNB1 mutation through Sanger sequencing. Using a separate cohort, the authors detect CTNNB1 32-37 mutations in urine cfDNA in sub-cohort of pre- and post-surgical specimens. The data and associations with recurrence outcomes are quite promising. There are, however, several limitations in the analysis, many of which are disclosed by the authors. More concerning is some omitted analysis that would substantially strengthen the impact of the manuscript. Specific concerns are listed below:
- Confirming urine CTNNB1 cfDNA mutations in cohort 1 would solidify the approach and analysis of cohort 2. Were urine specimens unavailable in cohort 1? Although the cohort description methods are lean, it appears the specimens were collected through a unified IRB. Were urine specimens only available in cohort 1? Was the HCC tissue in cohort 1 obtained post-resection? Was this same tissue available for analysis in cohort 2? A cross-confirmation (tissue x urine) in either cohort would substantially strengthen the conclusion of assay specificity.
- From the previous comment, the methods and cohort descriptions are quite lean. The authors should provide more detail on the staging and grading system utilized. Both cohorts are characterized by quite large (> 5cm) and aggressive HCC (assuming the cohort specimens with Stage 2 reflect vascular invasion). Since the urine cfDNA analysis represents a cohort within a cohort, the demographics of the patients with detectable urine cfDNA would be important to disclose. Also, the median follow-up (post-surgical) collection time is not reported.
- The recurrence outcomes in cohort 2 for HCC patients testing negative for CTNNB1 in the urine cfDNA should be reported.
- Although the authors acknowledge longitudinal follow-up as a limitation, confirming a positive CTNNB1 urine cfDNA specimen at the time of recurrence in patients not-detectable post-surgery would substantially strengthen the argument for surveillance in disease management.
- The conclusion of the potential for urine CTNNB1 32-37 cfDNA as a tool for precision medicine is not supported by the data and beyond the scope of the manuscript.
Author Response
Reviewer “…The data and associations with recurrence outcomes are quite promising. There are, however, several limitations in the analysis, many of which are disclosed by the authors. More concerning is some omitted analysis that would substantially strengthen the impact of the manuscript. Specific concerns are listed below:”
Point 1. Confirming urine CTNNB1 cfDNA mutations in cohort 1 would solidify the approach and analysis of cohort 2. Were urine specimens unavailable in cohort 1? Was the HCC tissue in cohort 1 obtained post-resection? Was this same tissue available for analysis in cohort 2? A cross-confirmation (tissue x urine) in either cohort would substantially strengthen the conclusion of assay specificity.
Response: Thank you for the suggestion and we agree. Unfortunately, cohort 1 is from archived tissue DNA and no urine was collected for this cohort. All HCC tissue DNA in cohort 1 were isolated from FFPE sections of surgically resected tissue, they are not post-resection tissue. We have included this information in the revised manuscript (lines 71-77). Since cohorts 1 and 2 are two independent cohorts, the tissue obtained from the cohort 1 cannot be used for cohort 2. We agree that a cross-confirmation (tissue x urine) would substantially strengthen the conclusion of the assay specificity. Unfortunately, we were not able to assess this due to sample availability. We have revised our conclusion in a more conservative manner in the revised manuscript (line 248)
Point 2. From the previous comment, the methods and cohort descriptions are quite lean. The authors should provide more detail on the staging and grading system utilized Both cohorts are characterized by quite large (> 5cm) and aggressive HCC (assuming the cohort specimens with Stage 2 reflect vascular invasion). Since the urine cfDNA analysis represents a cohort within a cohort, the demographics of the patients with detectable urine cfDNA would be important to disclose. Also, the median follow-up (post-surgical) collection time is not reported
Response: The staging is by the AJCC (TNM) tumor staging. All specimens were collected at National Cheng Kung University Hospital and are of Asian ethnicity. The tumor sizes are in a range of 1.5-8 cm and 46 of the 62 samples in cohort 2 are from early stages HCC, stage 1 & 2. We have now clarified this information in the Methods section and Tables 1-3.
Point 3. The recurrence outcomes in cohort 2 for HCC patients testing negative for CTNNB1 in the urine cfDNA should be reported.
Response: As HCC is highly heterogeneous, we did not know which circulating tumor DNA (ctDNA) marker to follow up when CTNNB1 mutations were not detectable before surgery. Undetectable CTNNB1 mutations in post-surgery urine could represent effective removal of tumor for a patient with CTNNB1 positive urine collected before surgery, but does not rule out minimum residual disease (MRD) or HCC recurrence arising from a new HCC nodule. So, it will be difficult to interpret the recurrence outcome from CTNNB1-negative patients. In this study CTNNB1 serves as a marker to demonstrate the potential of ctDNA in urine for detection of minimum residual disease. We therefore only focused on CTNNB1 marker positive patients to demonstrate the potential of urine ctDNA for detection of MRD.
Point 4. Although the authors acknowledge longitudinal follow-up as a limitation, confirming a positive CTNNB1 urine cfDNA specimen at the time of recurrence in patients not-detectable post-surgery would substantially strengthen the argument for surveillance in disease management.
Response: Agree, unfortunately, the initial study design was for a feasibility study to demonstrate that HCC ctDNA can be detected in urine. Urine was collected only before and after surgical resection. It was not designed for a HCC recurrence study. A prospective recurrence study is currently in progress and the specimens collected from the current on-going study will be able to strengthen the argument for surveillance in disease management as suggested by the reviewer. We hope the reviewer agrees that although there are limitations, the observation in this study is quite promising and will be of the interest to the readers of Diagnostics.
Point 5. The conclusion of the potential for urine CTNNB1 32-37 cfDNA as a tool for precision medicine is not supported by the data and beyond the scope of the manuscript.
Response: Thank you for the comment. We have revised the conclusion to be more specific based on our current evidence in the revised manuscript (line 248)

Reviewer 2 Report
In the manuscript the Authors discuss potential use of cfDNA from urine samples as a matherial for diagnostics of hepatocellular carcinoma. In detail, the beta-catenin gene (CTNNB1) in its mutational hotspot were analysed.
I think, that the data presented are interesting and well described. A short amplicon qPCR assay were used and initial results confirmed utility of cfDNA testing.
The major limitation is the small cohort of HCC positive samples. However, in Discussion the Authors reliably described the weaker side of the manuscript, e.g. stating that multi-center studies of the correlation between CTNNB1 mutations in urine cfDNA and HCC tumor status are needed to evaluate the potential clinical utility of urinary CTNNB1 mutation detection in liver cancer management and precision medicine.
Detailed comments:
The manuscript is well written.
In my opinion, sentence in line 140-141 should be corrected: abbreviation "VAF" should be described when used for the first time; precise meaning of "sensitivity" should be elucidated and the verb in the sentence is missing.
Author Response
Reviewer 2 “..the data presented are interesting and well described… The major limitation is the small cohort of HCC positive samples. However, in Discussion the Authors reliably described the weaker side of the manuscript, … The manuscript is well written. In my opinion, sentence in line 140-141 should be corrected: abbreviation "VAF" should be described when used for the first time; precise meaning of "sensitivity" should be elucidated and the verb in the sentence is missing.
Response: We thank Reviewer 2 for the recognition and comments. We have revised the sentence accordingly in the revised manuscript (lines 147-149).

Round 2
Reviewer 1 Report
Thank you for the opportunity to review the manuscript by Lin et. al. The authors have addressed my concerns, some of which cannot be improved due to limitations in the study design.